# The Bio-Diversity and the Role of Gut Microbiota in Postmenopausal Women with Luminal Breast Cancer Treated with Aromatase Inhibitors: An Observational Cohort Study

**DOI:** 10.3390/pathogens11121421

**Published:** 2022-11-26

**Authors:** Angioletta Lasagna, Mara De Amici, Chiara Rossi, Valentina Zuccaro, Marta Corbella, Greta Petazzoni, Francesco Comandatore, Lucia Sacchi, Giorgia Testa, Elisa Ferraris, Gianpiero Rizzo, Richard Tancredi, Alessandra Ferrari, Marco Lucioni, Paolo Sacchi, Raffaele Bruno, Paolo Pedrazzoli

**Affiliations:** 1Medical Oncology Unit, Fondazione IRCCS Policlinico San Matteo, 27100 Pavia, Italy; 2Division of Infectious Diseases I, Fondazione IRCCS Policlinico San Matteo, 27100 Pavia, Italy; 3Department of Microbiology and Virology, Fondazione IRCCS Policlinico San Matteo, 27100 Pavia, Italy; 4Laboratory of Immuno-Allergology of Clinical Chemistry and Pediatric Clinic, Fondazione IRCCS Policlinico San Matteo, 27100 Pavia, Italy; 5Unit of Anatomic Pathology, Fondazione IRCCS Policlinico San Matteo, 27100 Pavia, Italy; 6Department of Biomedical and Clinical Sciences, Romeo and Enrica Invernizzi Pediatric Clinical Research Center, University of Milan, 20126 Milan, Italy; 7Laboratory for Biomedical Informatics “Mario Stefanelli”, Department of Electrical, Computer and Biomedical Engineering, University of Pavia, 27100 Pavia, Italy; 8Pediatrics Clinic, Fondazione IRCCS Policlinico San Matteo, University of Pavia, 27100 Pavia, Italy; 9Department of Clinical Surgical Diagnostic and Pediatric Sciences, University of Pavia, 27100 Pavia, Italy; 10Department of Internal Medicine and Medical Therapy, University of Pavia, 27100 Pavia, Italy

**Keywords:** estrobolome, microbiome, TILs, breast cancer, IL17, aromatase inhibitors, luminal breast cancer

## Abstract

The interactions between aromatase inhibitors (AI) in breast cancer (BC) and gut microbiota (GM) have not been completely established yet. The aim of the study is to evaluate the bio-diversity of GM and the relationship between GM, inflammation and tumor-infiltrating lymphocytes (TILs) in postmenopausal women with BC during adjuvant AI treatment compared to women with disease relapse during or after one year of AI therapy (“endocrine-resistant”). We conducted a monocenter observational case-control study. Eighty-four women with BC (8 cases, 76 controls) were enrolled from 2019 to 2021. We observed a significant difference in the mean microbial abundance between the two groups for the taxonomic rank of order (*p* 0.035) and family (*p* 0.029); specifically, the *case* group showed higher diversity than the *control* group. *Veillonella* reached its maximum abundance in *cases* (*p* 0.022). Cytokine levels were compared among the groups created considering the TILs levels. We obtained a statistically significant difference (*p* 0.045) in IL-17 levels among the groups, with patients with low TILs levels showing a higher median value for IL-17 (0.15 vs. 0.08 pg/mL). Further studies about the bio-diversity in women with BC may lead to the development of new biomarkers and targeted interventions.

## 1. Introduction

Breast cancer (BC) is the most frequent solid tumor worldwide for both sexes, and accumulating data report an increased incidence rate. In 2020, the estimated incidence of breast cancer was 2.3 million cases (11.7% of total cases) with about 684,000 estimated deaths [1]. In postmenopausal women, BC risk increases with the weight gain linked to high levels of endogenous estrogens, leading to the alteration of host inflammation and metabolism [2].

The microbiota is the amount of the living microorganisms associated to the human body and it is mainly composed of bacteria, with a minor component of fungi, archeae and viruses [3]. It is implicated in the development of various types of cancer [4] and involved in many aspects of tumor biology, such as innate and cell-mediated immunity and hormone availability [5]. Several studies have investigated the relationship between gut microbiota (GM) dysbiosis and BC. GM seems to be able to modulate the serum levels of estrogens: in particular, Plottel and Blaser have defined the amount of enteric bacterial genes whose products are able to metabolize estrogens and their metabolites, and to modulate the enterohepatic circulation of estrogens as the “estrobolome” [6]. In particular, the estrobolome consists of some bacterial genes encoding β−glucuronidase and/or β−galactosidase that regulate the estrogen metabolism in the human body [7]. A recent paper found that bacteria producing β-glucuronidase typically included *Collinsella* and *Edwardsiella*; bacteria producing β-galactosidase included *Dorea*, *Klebsiella* and *Staphylococcus*; bacteria producing both the metabolites included *Alistipes*, *Bacteroides*, *Bifidobacterium*, *Faecalibacterium*, *Lactobacillus* and *Roseburia* [8]. An estrobolome enriched in bacterial genes encoding β−glucuronidase and/or β−galactosidase might lead to greater relative levels of circulating free estrogens [9]. In postmenopausal women, the adipose tissue is able to product estrogen by the aromatization of androgen precursors. Aromatase inhibitors (AI) potently inhibit the aromatase activity and suppress estrogen levels in plasma and tissue [10]. GM composition and biodiversity influences the regulation of the various types of hormones, but the dysbiosis, defined as “the abnormal composition of the microbiome” [11], seems to be associated with postmenopausal but not premenopausal BC [12] and the interactions between AI and GM has not completely established yet.

Moreover, GM might promote malignancy by inducing chronic inflammation and by triggering uncontrolled innate and adaptive immune responses. Increased BC risk has been associated with the presence of chronic and dysregulated inflammation [13]. For instance, one postulated inflammation-related mechanism for breast cancer is the up-regulation of cyclooxygenase 2 (COX2) and its product, prostaglandin E2 (PGE), with a consequent increased aromatase expression in adipose tissue and conversion of androgen precursors to estrogens [14,15]. Prebiotics are able to restore a dysbiosis. For example, prebiotics such as enterolactone can enhance the growth and activity of beneficial gut microorganisms, such as phytoestrogens, that work as antioxidants causing downregulation of COX2-mediated inflammation [5].

Finally, a recent published study by Shi and colleagues showed a correlation between the diversity of the gastrointestinal microbiome and the presence of tumor-infiltrating lymphocytes (TILs) in patients with BC [16]. Abundant TILs are associated with better outcome for patients with triple negative BC and HER2-positive BC, while the prognostic significance of TILs in estrogen receptor (ER)-positive/HER2-negative BC remains unclear [17]. Furthermore, the relationship between GM, inflammation and TILs in patients with BC during AI has not already been investigated.

The aim of this study is to evaluate the GM compositions, inflammation and TILs in post postmenopausal women with BC during adjuvant AI treatment compared to women with disease relapse during or after 1 year of AI therapy (“endocrine-resistant”).

## 2. Materials and Methods

### 2.1. Study Setting

We evaluated postmenopausal women with ER/progesterone (PgR)-positive and HER2-negative BC (with or without previous anthracycline- and taxane-based adjuvant chemotherapy) undergoing adjuvant hormonal treatment with AI (anastrozole, letrozole, exemestane) since at least three years therapy at the Medical Oncology Unit of Fondazione IRCCS Policlinico San Matteo Pavia. The study population was split into *cases* (patients who documented disease relapse during AI therapy or within 12 months of completing adjuvant AI) and *controls* (patients who did not experience relapse). The study period ran from November 2019 to July 2021. The exclusion criteria were the following: the onset of menopause after BC diagnosis (including perimenopause period); a personal history of other type of cancer (with exception for non-melanoma skin cancer); previous treatment with tamoxifen and with an LHRH analogue; previous adjuvant chemotherapy different from the sequential treatment with anthracycline and taxanes; patients reporting an intake of antibiotic therapy during the last 3 months; a personal history of autoimmune or inflammatory bowel disease; presence of gastrointestinal symptoms suggestive for colitis (such as diarrhea, abdominal pain); any major intestinal surgery (including bariatric surgery) in the previous six months.

The study was conducted according to the Strengthening the Reporting of Observational Studies in Epidemiology (STROBE) Statement for reporting observational studies [18] and was approved by the local Ethics Committee (Comitato Etico Area Pavia) and Institutional Review Board (P-20190073421). All the subjects signed an informed written consent form.

Figure 1 highlights the main characteristics of this study.

### 2.2. Outcomes

The primary outcome of this study was to assess whether the GM biodiversity differed between endocrine-resistant cases and endocrine-sensitive controls. Then, we compared the characteristic of *cases* and *controls* in terms of clinical oncological medical history (disease stage at the time of diagnosis, histotype, hormonal receptor status, proliferation index, HER2, type of surgery, radiation therapy and kind of adjuvant therapy) and GM composition. Secondary outcomes were to compare TILs levels and serum marker of immune-activation, immune-exhaustion and bacterial translocation among *cases* and *controls.* Moreover, we compared the characteristic of GM according to clinicopathological features between *cases* and *controls*.

### 2.3. Biological Samples

The samples (stool and blood) were obtained from the patients at the fifth year of the adjuvant endocrine therapy (*controls*) or at the time of the relapse during AI therapy or within 12 months of completing adjuvant AI (*cases*).

#### 2.3.1. Stool Sample Processing and DNA Extraction

Stool samples were kept at −80 °C, at the Laboratory of Microbiology and Virology of IRCCS Foundation Policlinico San Matteo of Pavia.

Genomic DNA was extracted from fecal samples using QIAamp^®^ Fast DNA Stool Mini kit (Qiagen, Hilden, DE, USA) according to the manufacturer’s instructions. The DNA concentration of each sample was assessed using a Qubit 3 fluorometer (Invitrogen, Carlsbad, CA, USA). The V3–V4 hypervariable regions of the 16S rRNA gene were targeted for amplicon production and sequencing was conducted by BMR Genomics Srl (Padova, Italy) using a Paired-End, 2 × 300-bp cycle run on Illumina MiSeq sequencing system.

#### 2.3.2. Blood Sample Processing and DNA Extraction

Blood samples were taken from an antecubital vein of the forearm of each subject, after overnight fasting. The serum was centrifuged and then frozen at −20°C for the subsequent analysis.

The human Interleukin 17 (IL-17) immunoassay (Human IL17 Immunoassay, R&D Systems, Minneapolis, MN) employs the quantitative sandwich enzyme immunoassay technique and was performed according to the manufacturer’s instructions and expressed in pg/mL. The minimum detectable dose (MDD) of IL-17 was less 15 pg/mL (determined by adding two standard deviations to the mean optical density value of twenty zero standard replicates and calculating the corresponding concentration).

The human Interleukin CD14 (CD14) immunoassay (Human CD14 Immunoassay, R&D Systems, Minneapolis, MN) employs the quantitative sandwich enzyme immunoassay technique and was performed according to the manufacturer’s instructions and expressed in pg/mL. The MDD of CD14 was less 125 pg/mL (determined by adding two standard deviations to the mean optical density value of twenty zero standard replicates and calculating the corresponding concentration).

The human Interleukin prostaglandin E2 (PGE2) immunoassay (Human PGE2 Immunoassay, R&D Systems, Minneapolis, MN) employs the quantitative sandwich enzyme immunoassay technique and was performed according to the manufacturer’s instructions and expressed in pg/mL. The MDD of PGE2 was from 16.0–41.4 pg/mL (determined by adding two standard deviations to the mean optical density value of twenty zero standard replicates and calculating the corresponding concentration) and the mean MDD was 30.9 pg/mL. IL17 and CD14 serum titers were evaluated in the peripheral blood of all the above time of the patients.

### 2.4. TILs Levels

TILs levels in breast cancer tissues were evaluated on hematoxylin and eosin (HE) stained sections according to the 2014 International Immuno-Oncology Biomarker Working Group on Breast Cancer [19]. In brief, TILs were assessed as the percentage of the stromal tissue within the borders of the invasive component alone that was occupied by mononucleated inflammatory cells. The average TILs percentage was rendered, avoiding hotspots, and it was scored as a continuous variable (0–100%). The values thus obtained were then used to categorize the tumor into three categories: low TILs (0–9% of stromal TILs), intermediate TILs (10–49% of stromal TILs) and lymphocyte-predominant BC (≥50% of stromal TILs).

### 2.5. Statistical and Bioinformatic Analyses

Raw reads were processed using an ad-hoc bioinformatics pipeline (Arrow Diagnostics) built under the R environment and the Microbiome Analyst v. 3.5.1 (www.microbiomeanalyst.ca accessed on 4 July 2022) online tool. Operational taxonomic units (OTUs) were classified at 97% homology level after filtering for sequences not passing the quality control. Taxonomy was then assigned against the Ribosomal Database Project (RDP) reference database, release 11. Low-count (20% prevalence cut-off) and low-variance (based on the inter-quartile range) filters were applied, and data rarefaction and scaling (through total sum normalization) were performed (microbiome analyst default parameters). Finally, filtered OTUs were used to compute relative abundances of microbial taxa in each sample. Microbial profiles of taxa with at least prevalence > 5% in one sample of the dataset were compared between patient groups (*cases* and *controls*) using the Mann–Whitney U-test. Significance threshold (*p*-value) was set to 0.05.

Quantitative variables were summarized as the median and interquartile range (IQR), and categorical variables were summarized with frequencies.

The α-diversity indexes (observed richness and Shannon) were computed at all taxonomic levels to analyze the within-sample diversity. Results were compared between groups using the Mann–Whitney U-test (*p*-value threshold set at 0.05).

The β-diversity (diversity in composition among samples) was computed at all taxonomic levels. Dissimilarity matrices were calculated using the Bray–Curtis distance method and visualized as principal coordinates analysis (PCoA). Permutational multivariate analysis of variance (PERMANOVA) was then performed for β–diversity analysis to assess the grouping of samples.

Predictive modeling was performed using random forest and gradient boosting. Given the low number of samples available, models were evaluated using a leave-one-out procedure.

Data analysis was performed using MatlabR202b (The Mathworks, Inc.) and the Orange Data Mining suite [20].

## 3. Results

A total of 84 women with BC (8 cases, 76 controls) were enrolled from November 2019 to July 2021. Overall, twelve BC patients (14%) had an invasive lobular carcinoma (ILC), one patient had mucinous carcinoma (1%) while seventy-one (84%) had invasive ductal carcinoma (IDC). In Table 1 we have reported the main characteristics according to cases and controls. According to the 13th St Gallen International Breast Cancer Conference [21], six cases (75%) and forty-nine controls (65%) presented a Luminal A-like subtype, while two cases (25%) and twenty-seven controls (35%) presented a Luminal B-like subtype. Histological grade was assessed with the Notthingham Histologic Score: nine patients (all controls) were graded as G1, fifty controls and six cases were graded as G2, seventeen controls and two cases were graded as G3. Twenty-two patients had received adjuvant chemotherapy (antracycline and taxane-based): nineteen controls (25%) and three cases (37%). With regard to the type of AI, seventy-four controls (97%) and two cases (25%) were receiving anastrozole at the time of study enrollment. When stratified by body mass index (BMI), thirty-six controls (47%) and four cases (50%) presented overweight (BMI 25–30 kg/m^2^), four controls (6%) had lower BMI and thirty-six controls (47%) and four cases (50%) were in the normal range.

### 3.1. Taxonomic Structure of Fecal Bacterial Communities in Cases and Controls

We investigated shifts in structure and composition of fecal bacterial communities across cases and controls. No statistically significant difference was found in average relative abundance for the most represented phyla, class and order in *cases* and *controls* as shown in Figure 2a–d. *Firmicutes* and *Bacteroidetes* were the most abundant in cases and controls (Figure 2a). Specifically, *Firmicutes* frequencies were 69.4% in the case group and 70.8% in controls as *Bacteroidetes* were 15.5% and 16.8%, respectively. Concerning the taxonomic rank of class, the most abundant both in cases and controls were *Clostridia* (60.6% and 63.6%, respectively), and *Bacteroidia* (14.6% and 16.9%, respectively) (Figure 2b). With regards of order, *Clostridiales* were the most abundant in all groups: 61.0% in cases and 64.2% in controls (Figure 2c).

As to family, *Veillonella* reached its maximum abundance in *cases* (*p* 0.022) as shown in Figure 3.

The study population was later split according to TILs levels into three categories: low TILs (LT), intermediate TILs (IT) and lymphocyte-predominant BC. Given the lack of patients in the latter category, the fecal bacterial communities were investigated across low TILs and intermediate TILs groups. As shown in Figure 4, we did not find any statistically significant difference in average relative abundance for the most represented phyla, class and order. (Figure 4). Specifically, *Firmicutes* frequencies were 70.5% in LT and 71.2% IT as *Bacteroidetes* were 16.7% and 17.0%, respectively. Concerning the taxonomic rank of class, the most abundant both in LT and in IT were *Clostridia* (61.7% and 64.9%, respectively), and *Bacteroidia* (16.4% and 16.4%, respectively) (Figure 4b). With regards of order, *Clostridiales* were the most abundant in all groups: 61.7% in LT and 63.9 % in IT (Figure 4c).

Furthermore, we categorized the study population according to luminal subtype (luminal A subtype (LA) and Luminal subtype B (LB)). Then, we compared the bacterial communities between LA and LB, both in *cases* and *controls*. As shown in Figure 5, no statistically significant difference was found in average relative abundance for the most represented phyla, class and order. *Firmicutes* and *Bacteroidetes* were the most abundant in subtype Luminal A and B in both *cases* and *controls* (Figure 5a). Specifically, in the *case* group *Firmicutes* frequencies were 69.7% for LA and 70.3% for LB while *Bacteroidetes* were 15.5% and 16.4%, respectively. Only one patient with LA showed higher abundance of *Actinobacteria* than *Bacteroides*. In the *control* group, *Firmicutes* and *Bacteroidetes* frequencies were 71.9% and 16.1% for LA and 68.2% and 18.3% for LB. Concerning the taxonomic rank of class, in the *case* group, the most abundant were *Clostridia* in both subtypes (62.2% for LA and 51.7% for LB) and *Bacteroidia* (15.8% for LA and 14.1%,for LB) (Figure 5b). In the *control* group, *Clostridia* and *Bacteroidia* frequencies were 63.0% and 15.8% for LA and 61.8% and 18.2% for LB. With regards of order, *Clostridiales* were the most abundant in all groups; specifically in the *case* group, the frequencies for LA and LB were 62.2 % and 51.7%, respectively, and 63.0% and 61.7% in the *control* group (Figure 5c).

### 3.2. Ecological Analyses of Fecal Communities in Cases and Controls

The within-sample diversity was evaluated by α-diversity indexes (observed richness and Shannon). Results were compared between groups using the Mann–Whitney U-test (*p*-value threshold set at 0.05). We calculated Chao1 indices by Wilcoxon rank-sum test and we observed a significant difference in the mean microbial abundance between two groups for the taxonomic rank of order (*p* 0.035) and family (*p* 0.029), specifically, the case group showed higher diversity than the control group. Even though there was no significant difference, we observed an overall trend of decreasing richness also for the taxonomic rank of phylum and class. Moreover, we calculated Shannon index, which represents the observed number of species in the two groups, and we did not find a significant difference among the case and control group (Figure 6).

α-diversity indexes were computed also according to TILs categorization. We calculated Chao1 index by Wilcoxon rank-sum test and Shannon index. We did not observe any significant difference in the mean microbial abundance between LT and IT groups for the taxonomic rank of phylum, order and family (Figure 7).

Similarly, we performed within-sample diversity evaluation by α-diversity indexes in the study population split according to luminal subtypes (A and B). Concerning the case group, Chao1 index by Wilcoxon rank-sum test and Shannon index did not show any significant difference between the LA and LB group. Although the sample is small and there is no statistically significant difference, we observed an overall trend of decreasing richness for the taxonomic rank of phylum, class and order.

With regards to control, Chao1 index did not show any difference between LA and LB. Meanwhile, we observed a significant number of species between LA and LB calculated by Shannon index for the taxonomic rank of Phylum (Figure 8).

The between-sample diversity (β-diversity) was evaluated by computing the Bray–Curtis dissimilarity matrix and was visualized through PCoA. Looking into the bacterial composition profiles, we did not observe significant differences between the case group and the control group (Figure 9). The analyses suggested no significant differences of all bacterial consortia as well when comparing the samples according to TILs and luminal subtype categorization (see Appendix A).

### 3.3. TILs Levels and Cytokine Levels

Of the 84 breast cancer specimens, 53 (63%, median value = 5%, range 1–5%) were classified as low TILs, and 20 (24%, median value = 13%, range 10–40%) were classified as intermediate TILs; no patients were found to belong to the lymphocyte-predominant group. In 11 cases, tumor sections without biopsy site were not available, meaning the quantification of TILs was not recommended according to the 2014 International Immuno-Oncology Biomarker Working Group on Breast Cancer recommendations [19].

Cytokine levels (IL17, CD14 and PGE) were compared among the three groups created considering the TILs levels (overall, low and intermediate). We obtained a statistically significant difference (*p* = 0.04) in IL-17 levels among the groups, with patients with low TILs levels showing a higher median value for IL-17 (0.15 vs. 0.08 pg/mL). Considering PGE and CD14, we did not obtain a statistically significant difference (*p* = 0.93 and *p* = 0.69, respectively) (Table 2).

Cytokine levels were compared between the cases and the controls. We did not obtain a statistically significant difference considering all three types of cytokines (Table 3).

We trained two machine learning models, random forests and gradient boosting, to predict TILs levels by using features related to the level 3 taxonomy of the microbiome. Classification performance was only slightly better than the majority classifier.

## 4. Discussion

In our cohort of patients with ER/PgR-positive and HER2-negative BC undergoing adjuvant hormonal treatment with AI, we demonstrated poor differences, in all the bacterial consortia, between patients who documented disease relapse during AI (cases) and patients who did not experienced relapse (controls). However, according to the literature data, fecal microbiota of postmenopausal women with breast cancer had elevated levels of *Clostridiaceae* [22].

The cross-talk between sex hormones and gut microbiota has emerged over the last few years, and the effect of the GM compositions on the levels of estrogens and their metabolites has been deeply investigated [23]. Recently Feng and colleagues described potential new approaches to prevent and/or treat BC by modulating GM [24].

To the best of our knowledge, no prior studies have explored the link between response to hormone therapies and the bio-diversity of the GM. The estrobolome, as well as a subset of beta-glucuronidase-producing gut bacteria, influences the level of estrogens. Ervin and colleagues demonstrated in vitro the ability of the gut microbial β-glucuronidase (GUS) enzymes to reactivate estrogens from their inactive glucuronides [7]. The gut bacteria-possessing ß-glucuronidases are capable of metabolizing estrogens and are essential to the enterohepatic circulation of the estrogens [9]. *Clostridia*, *Ruminococcaceae* and *Escherichia bacteria* typically produce beta-glucuronidase [25] and so GM may have an impact on estrogen signaling.

Therefore, we investigated the relative abundance of aforementioned genus-producing GUS enzymes between the case and control group. Our results did not demonstrate any significant difference. However, consistently with the literature data, we found that *Clostridia* was the most abundant class and *Clostridiales* was the most abundant order in the cases group. Moreover, the *Veillonella* family was the most abundant in our case group: it is able of producing the aforementioned enzymes leading to increasing levels of free estrogens [26] and in a recent study, it was assumed to survive well in a pro-inflammatory environment [27]. When we compared the bacterial communities according to luminal subtypes in the case and control group, we did not observe a statistically significant difference: *Firmicutes* and *Bacteroidetes* were the most abundant in subtype Luminal A and B both in cases and in controls.

Concurrently, the hormone therapy could influence estrogen-related metabolism through the estrabolome [28]. In 2018, Zhu et al. demonstrated that the fecal microbiota of postmenopausal women with breast cancer was characterized by a higher number of species when compared with postmenopausal controls [12]. On the contrary, Goedert et al. demonstrated that the fecal microbiota of postmenopausal women with breast cancer, compared with control patients, had statistically significantly lower alpha diversity (*p* ≤ 0.004), except for Shannon index [22].

To date, few studies about the role of GM in BC patients are ongoing. There are also three intervention trials (NCT04139993, NCT03358511, NCT03290651) with the aim to evaluate the systemic immunomodulatory effects of microbiota-based formulation [29]. Moreover, in a previous paper, our team hypothesized that the estrobolome might alter the susceptibility to COVID-19 by modulating the levels of estrogen and cytokines [30].

In our current study, we also assessed circulating cytokines as serum markers of immune-activation and bacterial translocation among the cases and control, and in relationship with TILs levels in the primary cancer. An increasing number of studies have established a relationship between cancer and inflammation. IL-17 is a pro-inflammatory cytokine produced by T helper 17 lymphocytes, and it promotes the proliferation, invasion and metastasis of BC cells [31]. It has direct and indirect effects in tumor cells and may modulate the drug sensitivity of cancer cells [32]. Our results revealed no statistically significant difference between cases and controls, but this might be due to the small sample size. Interestingly, we found a statistically significant correlation between IL-17 and low TILs. The prognostic role of the immune status of TILs in triple negative and HER2-positive subtypes has been extensively studied [33], while in luminal breast cancers (LBC) it remains more elusive [34]. Studies about the potential clinical relevance of TILs in LBC had reported conflicting findings [35,36]. Our study did not have the statistical power to allow a prognostic role for TILs to be determined, but future analyses will focus on the biodiversity of the GM in relationship to TILs.

Our study has several limitations need to be considered. First, there are several confounding variables that might influence the final interpretation of GM composition (such as BMI, type of diet and antibiotic drugs intake). Due to the small number of patients enrolled, it was not possible to fully balance these confounding factors. Second, this is a cross-sectional study and therefore only a stool and blood sample were collected at the recruitment time, lacking sampling at baseline. However, to the best of our knowledge, there has been no previous research on this topic.

## 5. Conclusions

In conclusion, we observed a significant difference in the mean microbial abundance between endocrine-resistant cases and endocrine-sensitive controls in terms of the taxonomic rank of order and family. Our study did not have the statistical power to allow a prognostic role for GM to be determined, but future analyses about the bio-diversity in women with BC may lead to the development of new biomarkers and targeted interventions.

## Figures and Tables

**Figure 1 pathogens-11-01421-f001:**
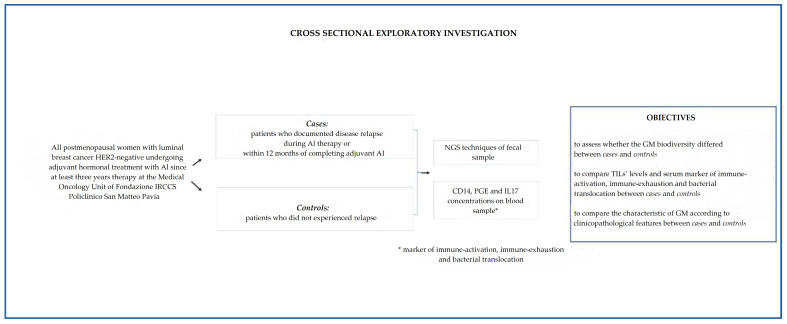
Cross-sectional exploratory investigation: materials and methods.

**Figure 2 pathogens-11-01421-f002:**
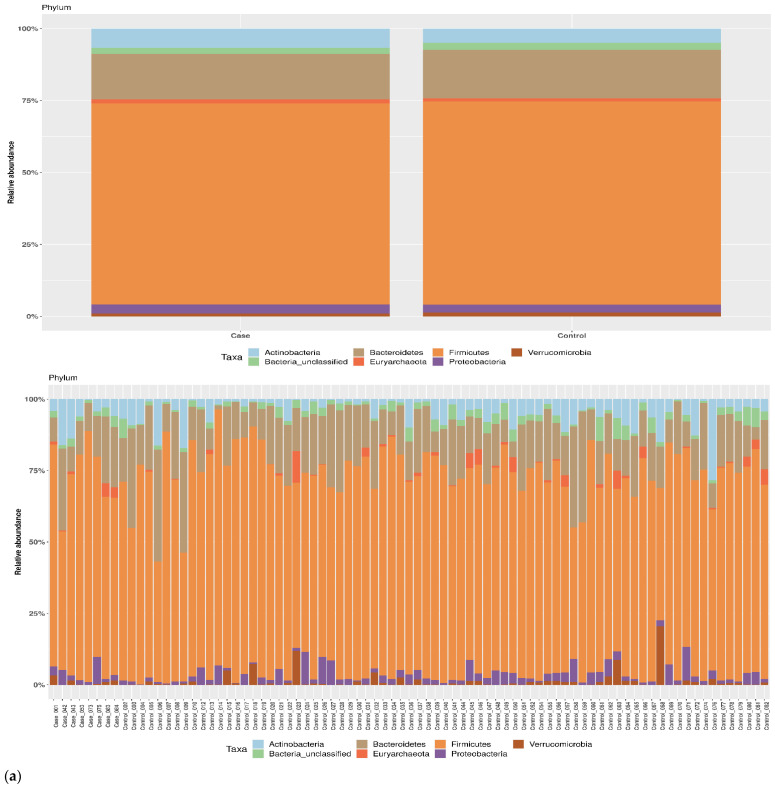
Taxonomic composition of the gut microbiota in cases and controls. Average relative abundances of the most represented phyla (**a**), class (**b**), order (**c**) and genus (**d**) identified in study groups. Only taxa whose relative abundance was >5% in at least one group were included.

**Figure 3 pathogens-11-01421-f003:**
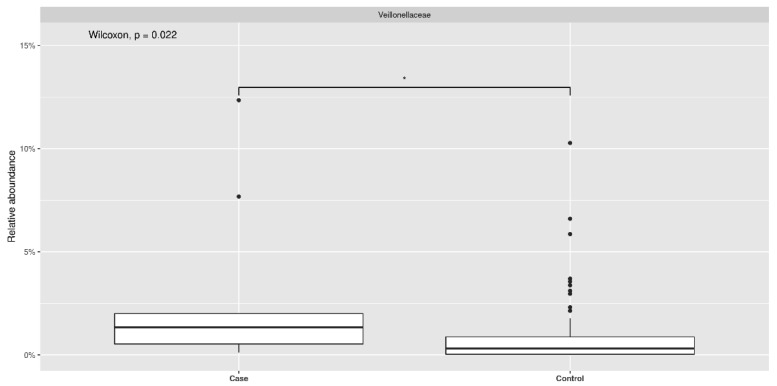
Distribution of relative abundance at the taxonomic level of family of *Veillonellaceace* in case and control group.

**Figure 4 pathogens-11-01421-f004:**
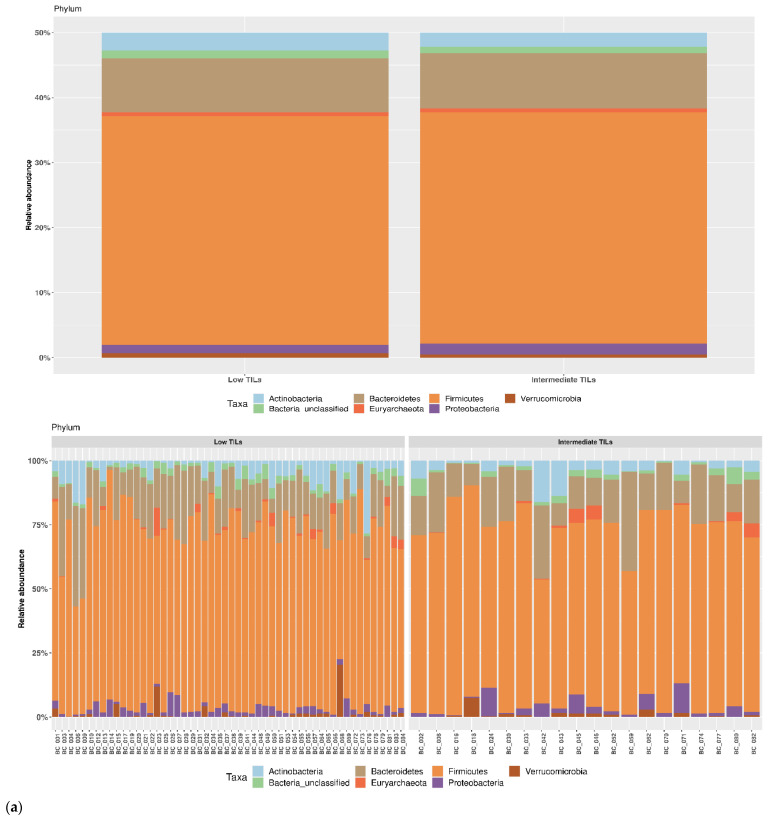
Taxonomic composition of the gut microbiota according to TILs levels. Average relative abundances of the most represented phyla (**a**), class (**b**), order (**c**) and genus (**d**) identified in low TILs (LT) and intermediate TILs. (IT) Only taxa whose relative abundance was >5% in at least one group were included.

**Figure 5 pathogens-11-01421-f005:**
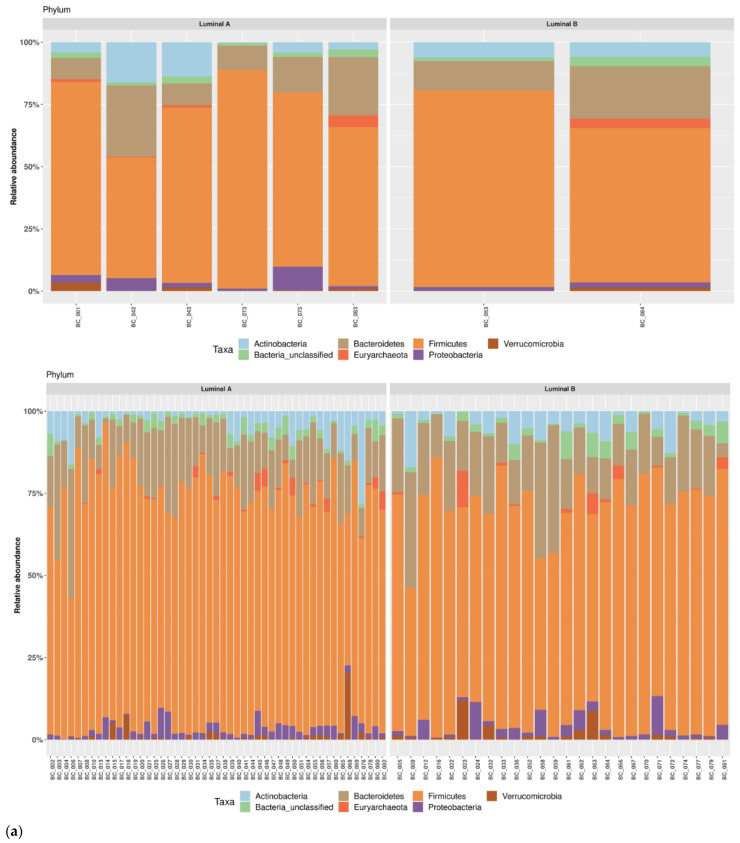
Taxonomic composition of the gut microbiota according to luminal subtypes. Average relative abundances of the most represented phyla (**a**), class (**b**), order (**c**) identified in Luminal A and Luminal B. Only taxa whose relative abundance was >5% in at least one group were included.

**Figure 6 pathogens-11-01421-f006:**
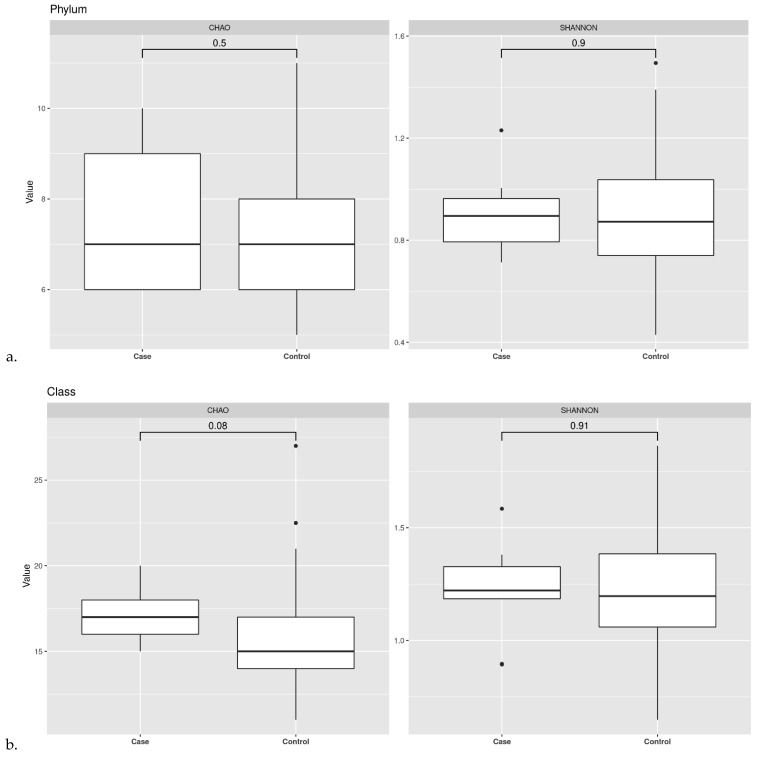
α-diversity. Observed richness and Shannon indices are presented at the taxonomic level of phylum (**a**), class (**b**), order (**c**) and family (**d**).

**Figure 7 pathogens-11-01421-f007:**
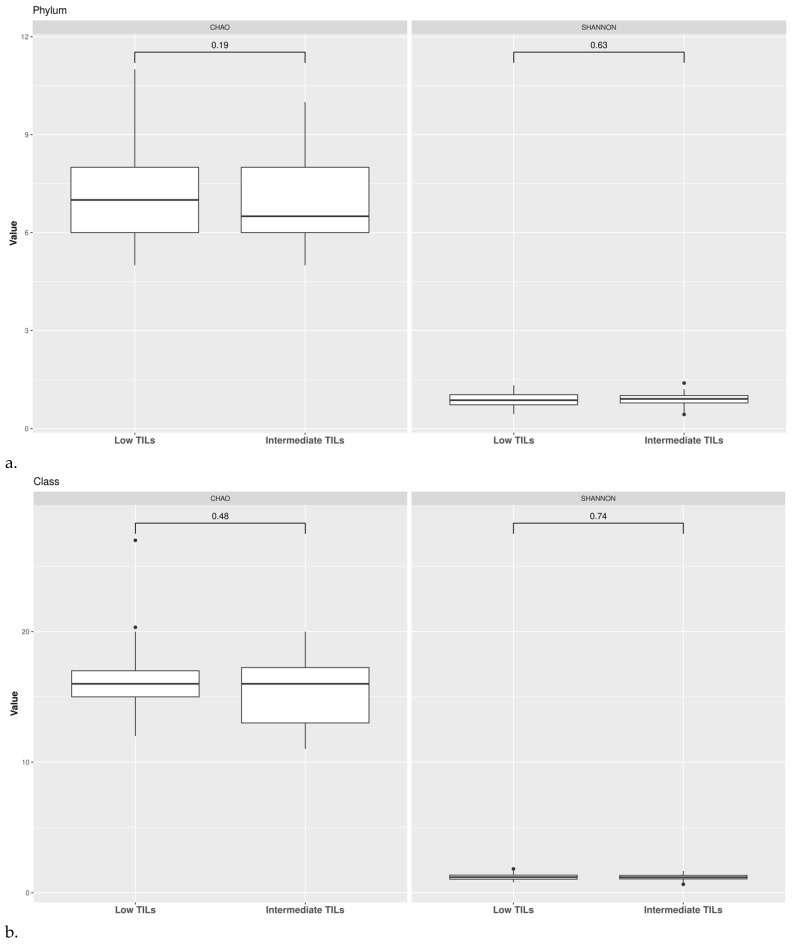
α-diversity according to TILs categorization. Observed richness and Shannon indices are presented at the taxonomic level of phylum (**a**), class (**b**), order (**c**).

**Figure 8 pathogens-11-01421-f008:**
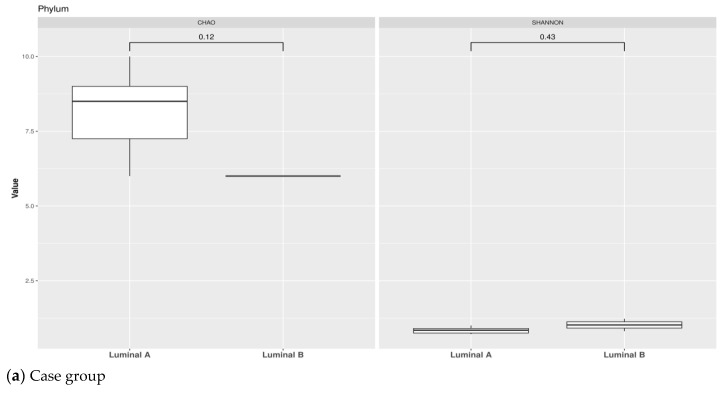
α-diversity according to Luminal subtypes. Observed richness and Shannon indices are presented at the taxonomic level of phylum in case group (**a**) and in control group (**b**); class in case group (**c**) and in control group (**d**); order in case group (**e**) and in control group (**f**).

**Figure 9 pathogens-11-01421-f009:**
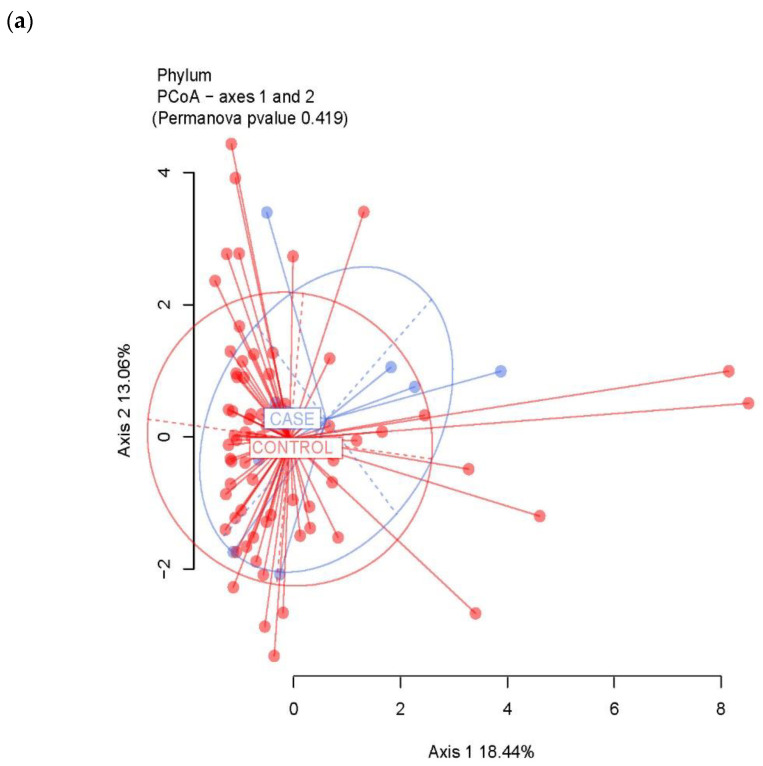
Beta-diversity. The microbiota distances were evaluated through the Bray–Curtis dissimilarity matrix at the taxonomic level of phylum (**a**), genus (**b**), order (**c**) and family (**d**) and visualized through Principal Coordinates Analysis (PCoA). Each point represents the microbiota composition of one sample.

**Table 1 pathogens-11-01421-t001:** Patients’ characteristics.

	Cases (*n* = 8)	Controls (*n* = 76)
Tumor histology		
IDC	8 (100%)	63 (83%)
ILC	0	12 (16%)
Other	0	1 (1%)
Tumor grade		
G1	0	9 (12%)
G2	6 (75%)	50 (66%)
G3	2 (25%)	17 (22%)
Estrogen receptor		
Positive	8 (100%)	76 (100%)
Negative	0	0
Progesterone receptor		
Positive	8 (100%)	76 (100%)
Negative	0	0
Ki-67		
Low	6 (75%)	60 (79%)
High	2 (25%)	16 (21%)
Molecular subtype		
Luminal A	6 (75%)	49 (65%)
Luminal B	2 (25%)	27 (35%)
TILs		
Low TILs	5 (63%)	48 (63%)
Intermediate TILs	2 (25%)	18 (24%)
Lymphocyte-predominant	0	0
nv	1 (12%)	10 (13%)
Adjuvant endocrine therapy		
Anastrozole	2 (25%)	74 (97%)
Letrozole	6 (75%)	2 (3%)
Exemestane	0	0
Adjuvant chemotherapy		
Yes	3 (37%)	19 (25%)
No	5 (63%)	57 (75%)
Comorbidities		
Diabetes mellitus	2 (25%)	8 (10%)
Autoimmune disorders	1 (12%)	2 (3%)
HCV	0	5 (6%)
HBV	0	1 (1%)
BMI		
25–29.9 kg/m^2^	4 (50%)	36 (47%)
18.5–24.9 kg/m^2^	4 (50%)	36 (47%)
>18.5 Kg/m^2^	0	4 (6%)

Abbreviations: ILC: invasive lobular carcinoma; IDC: invasive ductal carcinoma; BMI: body mass index; nv: not evaluated; HCV: Hepatits C virus; HBV: Hepatitis B virus.

**Table 2 pathogens-11-01421-t002:** TILs levels and cytokine levels.

	TILs’ Levels—All Patients (n = 84) (Median [IQR])	TILs’ Levels—Low (n = 53) (Median [IQR])	TILs’ Levels—Intermediate (n = 20) (Median [IQR])	*p*-Value (Low vs. Intermediate)
IL17	0.11 [0.04–0.21]	0.15 [0.04–0.24]	0.08 [0.02–0.11]	0.04
CD14	1828.50 [1081.1–2621.04]	1894.76 [1139.61—2552.24]	1433.70 [1049.86—2573.24]	0.69
PGE	1207.93 [672.33–1633.71]	1162.52 [672.33—1633.71]	1295.24 [599.40—1633.15]	0.93

Abbreviations: IL-17: human Interleukin 17; CD14: human Interleukin CD14; PGE2: prostaglandin E2; IQR: interquartile range.

**Table 3 pathogens-11-01421-t003:** TILs levels and cytokine levels between cases and controls.

	TILs Levels—All Patients (n = 84) (Median [IQR])	TILs Levels—Controls (n = 76) (Median [IQR])	TILs Levels—Cases (n = 8) (Median [IQR])	*p*-Value (Cases vs. Controls)
IL17	0.11 [0.04–0.21]	0.10 [0.03–0.21]	0.16 [0.07–0.28]	0.38
CD14	1828.50 [1081.1–2621.04]	1828.50 [1085.25–2664.73]	1947.49 [817.03–2232.15]	0.67
PGE	1207.93 [672.33–1633.71]	1221.90 [741.60–741.60]	591.46 [244.85—2600.24]	0.41

Abbreviations: IL-17: human Interleukin 17; CD14: human Interleukin CD14; PGE2: prostaglandin E2; IQR: interquartile range.

## Data Availability

All the data supporting the findings of this study can be found within the article.

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
