# Peer review of "The Bio-Diversity and the Role of Gut Microbiota in Postmenopausal Women with Luminal Breast Cancer Treated with Aromatase Inhibitors: An Observational Cohort Study"

_pathogens, 2022, doi:10.3390/pathogens11121421_

Round 1

Author Response

Reviewer: 1

Comments to the Author

Major changes

  1. Material and methods: a figure could be made that explains the study protocol, with

the objectives and inclusion criteria like those carried out for clinical trials. A figure of this type could facilitate the reading of the article.

Author response: We have added the figure 1, according to your suggestions.

  1. Results: since the study is performed on luminal breast tumors, I think it is essential to add the Ki67 of the patients in the study. An analysis divided by the luminal subtype (A and B) should be performed. Luminal breast cancers A are completely different from luminal B tumors, both in prognosis and treatment. Therefore, I think the results can be very different depending on this classification. Although the sample size is limited, it would not hurt to perform an analysis for this type of classification.

Author response: We have added the analysis according to this classification. We compared the bacterial communities between Luminal A and Luminal B, both in cases and controls. No statistically significant difference was found in average relative abundance for the most represented phyla, class and order.

α-diversity indexes were computed also according to luminal subtypes (A and B).

Moreover, we have evaluated the taxonomic composition of the gut microbiota, α-diversity and  β-diversity according to TILs levels (see figures 4,5,7,8)

  1. Results: together with the previous point, specify if the 8 cases were luminal patients

A or B. It stands to reason that these patients are mostly luminal B tumors.

Author response: We have added this information in the text and in the Table 1

Minor changes

  1. Title: it should be added that breast cancer is a luminal subtype. Today the different subtypes of breast cancer are completely different entities from each other, therefore, it should be specified in the title that the breast cancer studied is the luminal.

Author response: We have modified the title, according to your suggestions.

  1. Keywords: as in the title add "luminal breast cancer".

Author response: We have added the keyword "luminal breast cancer", according to your suggestions.

  1. Introduction: breast cancer is the most incident cancer worldwide for both sexes, and not only in women. Specify this information so that the reader understands the importance of the topic being addressed.

Author response: We have added this information in the text, according to your suggestions.

  1. References: I think it would be convenient to add the following references to complete the literature on the subject.

- Parida S, et al. Microbial Alterations and Risk Factors of Breast Cancer:

Connections and Mechanistic Insights. Cells. 2020 Apr 28;9(5):1091.

- Feng ZP, et al. Gut microbiota homeostasis restoration may become a novel

therapy for breast cancer. Invest New Drugs. 2021 Jun;39(3):871-878

Author response: We have added these references in the text, according to your suggestions.

Reviewer 2 Report

This manuscript presents an interesting and important study regarding gut microbiota (GM) in breast cancer patients treated with adjuvant aromatase inhibitors. The authors evaluated the GM compositions, TILs, and cytokine levels of breast cancer patients, and compared these factors between patients with and without recurrence. The study is interesting, and the manuscript is well written. However, there are several points that might be addressed to clarify the findings and improve the manuscript.   Major comments: 1. When were biological samples (stool and blood) obtained from the patients, before or after the initial treatment? 2. In Table 1, please provide age, tumor size, lymph node status, stage, estrogen receptor status, progesterone receptor status, and TILs as factors. Moreover, it might be better to compare each factor between the two groups (cases and controls). 3. Tables 2 and 3: Please present these data more clearly to make the results easier to understand. 4. Are there any clinicopathological factors, such as age, tumor histology, tumor size, etc., that are related to the composition of gut microbiota? Please compare the taxonomic composition of gut microbiota to each clinicopathological factor.   Minor comments: 1. Line 119; “HER 2” should be “HER2” 2. Figures 1, 2, and 3: Letters in the diagrams are too small. 3. “table 1”, “table 2”, and “table 3” in the text should be “Table 1”, “Table 2”, and “Table 3”.

Author Response

Reviewer: 2

Comments to the Author

Major comments:

  1. When were biological samples (stool and blood) obtained from the patients, before or after the initial treatment?

Author response: Samples (stool and blood) obtained from the patients at the fifth year of the adjuvant endocrine therapy (controls) or at the time of the relapse (cases).

We have added this information in the text.

  1. In Table 1, please provide age, tumor size, lymph node status, stage, estrogen receptor status, progesterone receptor status, and TILs as factors. Moreover, it might be better to compare each factor between the two groups (cases and controls).

Author response: We have modified the Table 1, according to your suggestions

  1. Tables 2 and 3: Please present these data more clearly to make the results easier to understand.

Author response: We have modified the tables, according to your suggestions

  1. Are there any clinicopathological factors, such as age, tumor histology, tumor size, etc., that are related to the composition of gut microbiota? Please compare the taxonomic composition of gut microbiota to each clinicopathological factor.

Author response: We have added the analysis according to the classification in Luminal A and Luminal B, both in cases and controls. Moreover, we have evaluated the taxonomic composition of the gut microbiota according to TILs levels (see figures 4,5,7,8)

Minor comments:

  1. Line 119; “HER 2” should be “HER2”

Author response: We have modified the text, according to your suggestions

  1. Figures 1, 2, and 3: Letters in the diagrams are too small.

Author response: We have modified the Figures, according to your suggestions

  1. “table 1”, “table 2”, and “table 3” in the text should be “Table 1”, “Table 2”, and “Table 3”.

Author response: We have modified the text, according to your suggestions

Round 2

Reviewer 1 Report

As I indicated in the previous review, I believe that the article written by the authors is of great value for publication in the journal if the indicated changes were made. The authors have made a great effort, and they have certainly made the indicated changes. There is currently no objection to the publication of the article.